# The Ameliorative Effect of Empagliflozin in Vigabatrin-Induced Cerebellar/Neurobehavioral Deficits: Targeting mTOR/AMPK/SIRT-1 Signaling Pathways

**DOI:** 10.3390/molecules27123659

**Published:** 2022-06-07

**Authors:** Rabab M. Amer, Amira Kamel Eltokhy, Rasha Osama Elesawy, Amany Nagy Barakat, Eman Basha, Omnia Safwat Eldeeb, Alshimaa Aboalsoud, Nancy Mohamed Elgharabawy, Radwa Ismail

**Affiliations:** 1Anatomy and Embryology Department, Faculty of Medicine, Tanta University, Tanta 31527, Egypt; rabab.amer@med.tanta.edu.eg (R.M.A.); radwa.ismaeil@med.tanta.edu.eg (R.I.); 2Medical Biochemistry Department, Faculty of Medicine, Tanta University, Tanta 31527, Egypt; omnia.eldeeb@med.tanta.edu.eg; 3Pharmacology Department, Faculty of Medicine, Tanta University, Tanta 31527, Egypt; rasha.elesawy@med.tanta.edu.eg (R.O.E.); alshimaa.taha@med.tanta.edu.eg (A.A.); 4Pediatric Department, Faculty of Medicine, Tanta University, Tanta 31527, Egypt; amany.barakat@med.tanta.edu.eg (A.N.B.); nancy.mohamed@med.tanta.edu.eg (N.M.E.); 5Physiology Department, Faculty of Medicine, Tanta University, Tanta 31527, Egypt; eman.basha@med.tanta.edu.eg

**Keywords:** vigabatrin, empagliflozin, GFAP, mTOR, AMPK, SIRT-1

## Abstract

**Introduction.** Vigabatrin (VGB) is an antiepileptic drug that acts to irreversibly inhibit the γ-aminobutyric acid (GABA) transaminase enzyme, elevating GABA levels. Broad studies have established that long-term treatment and/or high doses of VGB lead to variable visual defects. However, little attention has been paid to its other side effects, especially those demonstrating cerebellar involvement. Sodium glucose-linked co-transporter 2 (SGLT2) inhibitors are antidiabetic agents with protective effects far greater than expected based on their anti-hyperglycemic effect. **Method.** Our study herein was designed to investigate the possible ameliorative effect of empagliflozin, the SGLT2 inhibitors, in VGB-induced cerebellar toxicity. A total of 40 male Wistar rats were allocated equally into 4 groups: **Group I**: control group; **Group II**: VGB group; **Group III** empagliflozin treated VGB group; and **Group IV**: empagliflozin treated group. All groups were subjected to the detection of cerebellar messenger RNA gene expression of silent mating type information regulation 2 homolog 1 (SIRT1) and Nucleoporin p62 (P62). Mammalian target of rapamycin (mTOR), adenosine monophosphate-activated protein kinase (AMPK), and beclin1 levels were assessed by the ELISA technique while malondialdehyde (MDA) level and superoxide dismutase (SOD) activity were detected spectrophotometrically. Immuno-histochemical studies, focusing on glial fibrillary acidic protein (GFAP) and S100 were performed, and the optical color density and the mean area percentage of GFAP positive astrocytes and the number of S 100 positive cells were also counted. **Results.** Following empagliflozin treatment, we documented significant upregulation of both SIRT1 and P62 mRNA gene expression. Additionally, AMPK, Beclin1 levels, and SOD activity were significantly improved, while both mTOR and MDA levels were significantly reduced. **Conclusions.** We concluded for the first time that empagliflozin efficiently ameliorated the VGB-induced disrupted mTOR/AMPK/SIRT-1 signaling axis with subsequent improvement of the autophagy machinery and mitigation of the oxidative and inflammatory cellular environment, paving the way for an innovative therapeutic potential in managing VGB-induced neurotoxicity.

## 1. Introduction

Vigabatrin (VGB) is identified to be an antiepileptic drug that has established efficacy as a perfect treatment for refractory epileptic cases and for infantile spasms. Refractory epilepsies are known to have a strong deleterious impact on neurological integrity and life quality of the affected patients. VGB is considered as a structural analog of γ-aminobutyric acid (GABA) [1]. It was designed to enhance the function of γ-aminobutyric acid (GABA) in the CNS by inhibiting its catabolizing enzyme, gamma amino butyric acid transaminase (GABA-T) enzyme; thus, it elevates brain GABA level [2].

Vigabatrin (VGB) has an advantageous pharmacokinetic effect, as it does not undergo hepatic metabolism (does not stimulate the cytochrome p 450 system in the liver), is excreted by the kidney, has negligible protein binding capability, and has an extended efficient half-life, permitting once- or twice-daily dosing. Moreover, its interaction with other antiepileptic drugs is reported to be minimal [1].

Unfortunately, widespread studies have demonstrated that long-term treatment and high doses of VGB can cause variable visual field defects. However, little attention has been paid to its other side effects such as tremors, ataxia, and gait abnormalities, which indicate the involvement of the cerebellum [3]. Earlier studies reported Vigabatrin-induced cell loss and intra-myelinic edema in the cerebellum of rats [4,5], but the pathogenesis remains poorly understood.

The VGB-induced injury could be explained by GABA accumulation, which is toxic to the brain neurons and neuroglia. In fact, the accumulated GABA has a stimulatory effect on the mammalian target of rapamycin (mTOR)-signaling pathway along with the inhibition of adenosine monophosphate-activated protein kinase (AMPK) [6]. A dysregulated mTOR pathway is included in many pathological conditions including neurological disorders, metabolic diseases, epilepsy, and autism. There is a novel link between GABA metabolism, mTOR activity, and autophagy, as the stimulated mTOR has an inhibitory effect on autophagy pathways resulting in elevated mitochondrial numbers, dysregulated mitophagy, and pexophagy with associated oxidant stress (OS) as well as pro-inflammatory and pro-apoptotic status [7].

Both mTORC1 and AMPK act as key sensors for biochemical changes in cells; they regulate the activity of the protein components of autophagy, such as beclin1 and P62; while mTORC1 negatively regulates autophagy by phosphorylation and the inhibition of autophagy proteins, AMPK phosphorylates them at different positions and triggers autophagy. Thus, while mTOR activation ameliorates OS, AMPK protects cells from senescence-induced OS by restoring the autophagic machinery [8].

It is well known that SIRT1 can regulate redox status via NF-κB signaling, which induces the expression of various antioxidants, such as Mn-SOD and Zn-SOD. Moreover, there is complex integration between SIRT1 signaling and cellular autophagy, i.e., SIRT1 is a potent enhancer of autophagy. Interestingly, mTor and SIRT1 signaling have been documented to be reciprocally interplayed, exerting a neuroprotective cellular effect. An early study indicated that SRT1 has the ability to induce the growth and survival of neurons in the CNS through suppression of mTOR signaling, which, in turn, restores autophagic efflux [9]. Moreover, AMPK can activate SIRT1 by the induction of cellular NAD+ synthesis with subsequent restoration of the autophagic machinery of the cell [9].

Sodium glucose-linked cotransporter 2 (SGLT2) inhibitors are innovative antidiabetic agents that target SGLT2. Although these drugs are primarily approved for the treatment of diabetes, various recent studies have investigated their use in neurological diseases by virtue of their modulatory effect on the mTOR signaling pathway. SGLT2 inhibitors are drugs that are partly lipid-soluble and can cross the BBB. Numerous studies have established the presence of the SGLT2 receptor in numerous brain regions, comprising the hippocampus and cerebellum, which may highlight the impending therapeutic role of SGLT2 inhibitors in brain injury [10]. Notably, empagliflozin is highly selective for SGLT2 compared to dapagliflozin and canagliflozin, in addition, to having a high safety profile and very limited drug–drug interactions [11].

Therefore, **the aim of our study** is to investigate the potential effect of empagliflozin on the VGB-induced dysregulated mTOR signaling pathway and consequent impaired autophagy in cerebellar tissues of experimental rats.

## 2. Material and Methods

### 2.1. Drugs and Chemicals

Vigabatrin was purchased from Sigma-Aldrich (St. Louis, MO, USA) catalog number (720-929-6) while empagliflozin was obtained from Cayman Chemical (MI) catalog number (864070-44-0). The chemicals used, unless otherwise noted, were purchased from Sigma-Aldrich. All chemicals and solvents were of high analytical grade.

### 2.2. Animals

Forty adults male Wistar rats, weighted (150–200 g) aged 9 weeks, were included in this study. Animals were cared for according to Guidelines of the Care and Use of Laboratory Animals (1996, published by National Academy Press 2011, Washington, DC, USA). The experimental protocol was approved by the Animal Care Review Committee at the Faculty of Medicine, Tanta University. The rats were healthy and free from any disease or disability. They were maintained under standardized hygienic circumstances and given food and water ad libitum. They were allowed 7 days of acclimatization before the experiment.

### 2.3. Experimental Design

(1)**Group I (control group, n = 10)**, where the rats received a daily intraperitoneal (IP) injection of 0.9% sodium chloride along with oral administration of DMSO (1 mL/kg) by oral gavage for 6 consecutive weeks.(2)**Group II (vigabatrin group, n = 10)**, where the rats received an IP injection of 250 mg/kg/day vigabatrin dissolved in 0.9% sodium chloride (the final concentration was 25 mg in each one milliliter of 0.9% sodium chloride) for 6 consecutive weeks [4].(3)**Group III (Empagliflozin-treated vigabatrin group, n = 10)**, an IP injection of 250 mg/kg/day vigabatrin dissolved in 0.9% sodium chloride at a concentration of 25 mg/mL along with daily oral administration by oral gavage of 10 mg/kg/day empagliflozin dissolved in DMSO (the final concentration was 10 mg in each one milliliter of DMSO) starting on day 1 with the first dose of vigabatrin for 6 consecutive weeks [12].(4)**Group IV (Empagliflozin-treated group, n = 10)**, daily oral administration by oral gavage of 10 mg/kg/day empagliflozin dissolved in DMSO (the final concentration was 10 mg in each one milliliter of DMSO) starting on day 1 [12].

### 2.4. Behavioral Test

On the 43rd day, animals were subjected to behavioral/functional tests for evaluating locomotor activity (open-field test) and coordination (rotarod test).

#### 2.4.1. Open Field Activity

Open-field test was performed in a separated room with no interference noise or human activity. The locomotor activity was evaluated by means of an open-field box measuring 56 (long) × 42 (wide) × 40 cm (high) with the floor allocated into 12 squares. The number of squares crossed with the four paws was counted manually in a 6-min session by an observer who was blind to the animal condition. The number of crossed squares was used to analyze the locomotor activity [13].

#### 2.4.2. Rotarod Test

Rotarod test is commonly used to evaluate the cerebellar functions of rats via assessment of balance and motor coordination. In this test, the rat is put on a horizontally oriented, rotating rod suspended above a cage floor, which is low enough to avoid harming the animal, but high enough to prevent falling. Rodents naturally try to stay on the rotarod and avoid falling to the ground. Each rat was positioned on a rotarod (10 cm long and 4 cm in diameter). The rod rotation speed was 20 rpm. Each rat was left on the rotating rod and the time elapsed from putting the animal on the shaft of the rotarod, till it falls to the ground (latency) was recorded manually by an observer who was blind to the animal’s condition. The maximal time that was allowed for each rat was 2 min [14].

### 2.5. Data and Sample Collection

On the 44th day, the rats of the three groups were sacrificed. A midline incision was performed, then the brain was extracted carefully, and cerebellum was separated. The cerebellar hemispheres were dissected by a sagittal cut through the vermis.

### 2.6. Tissue Samples

One cerebellar hemisphere from each animal was fixed in 10% neutral-buffered formalin and used for histopathological and immunohistochemical assessments.

### 2.7. Histological and Immunohistochemical Assessments

The specimens were fixed in 10% neutral-buffered formalin and processed for paraffin blocks and prepared for light microscopic examination with Hematoxylin and Eosin stains (H&E) [15] and histopathological assessment.

For immunohistochemical staining, the 5 μm paraffin sections were dewaxed in an oven at 70 °C and then placed in xylene immersion (two baths for 2 min each), absolute ethanol (three baths 2 min each), then distilled water to rehydrate. The endogenous peroxidases were quenched with hydrogen peroxide (5%) in methyl alcohol for 10 min (3 times), protected from light. Non-specific crosslinks were blocked with 1% BSA (bovine serum albumin in PBS pH 7.4 Triton X-100, 0.05%) (1 h at room temperature). Then, the primary antibodies, anti-GFAP (monoclonal mouse anti-human, Clone: 6F2− DAKO, M0761, 1:500), and anti-S100 (polyclonal rabbit, DAKO Z0311, 1: 2500) were applied to each sample and initially incubated for 1 h at room temperature, followed by over-night incubation at 4 °C. Subsequently, the slices were washed with PBS for 5 min (3 times) and the samples for GFAP and S100 were incubated with secondary and tertiary monoclonal antibody Advance HRP Kit (DAKO) for 40 min each, washing with PBS-tx between each incubation. Finally, immunohistochemical reactivity was revealed with a 0.06% 3.3′-diami nobenzidine (DAB) (Dako) in PBS-tx for 3 min for GFAP and S100. The samples for S100 were counterstained with hematoxylin. Finally, the slides were dehydrated in ethanol at increasing concentrations (70, 90, and 100% for 2 min) and xylene. Subsequently, the slides were cover-slipped and examined [16].

Cerebellar sections were examined and photographed using Olympus BX 50 Automated light microscope in low and high magnifications. The images were digitized using an Olympus digital video camera (model NO. E-330 DC 7.4v)

The other hemisphere was homogenized in a twice volume of ice-cold TBS (50 mM TRIS, 150 mM NaCl, pH 7.4), the homogenates were centrifuged at 10.000× *g* for 15 min at 4oC, and the resultant supernatant was frozen at −20 °C for biochemical parameters analysis:

### 2.8. Biochemical Parameters Assessment

#### 2.8.1. Redox Status Parameters

**Superoxide dismutase (SOD) activity:** SOD activity was assessed by commercial kit (Biodiagnostic, Worcestershire, UK). Briefly, 100 µL of tissue homogenate was mixed with 1000 µL of working reagent and the reaction was initiated by adding 100 µL R4 to each sample. The increase in absorbance was read at 505 nm over 5 min [17].

**Malonaldehyde (MDA**): Level was determined depending on the formation of MDA as an end product of lipid peroxidation when lipid reacts with thiobarbituric acid and produces thiobarbituric acid reactive substance (TBARS), a pink chromogen, which can be assessed by spectrophotometry at 532 nm [18].

#### 2.8.2. Immunoassay of mTOR, Beclin1, and AMPK

Enzyme-linked immunosorbent assay (ELISA) was used to detect tissue m.TOR level (Sigma, St Louis, MO, USA), tissue **Beclin1** levels were assayed by ELISA kit supplied by (Glory Science Co., Hou-Ko, Road Hou-Li District Changhua Hsien, Taiwan), and **AMPK** levels (Glory Science Co., Hou-Ko, Road Hou-Li District Changhua Hsien, Taiwan) according to instructions of the manufacturer.

#### 2.8.3. Quantitative Analysis of SIRT1 and P62 Gene Expression by Quantitative Real-Time Polymerase Chain Reaction (RT-PCR)

(a)Firstly, RNA was extracted from tissue homogenate using Gene JET RNA Purification Kit (Thermo Fisher Scientific, Waltham, MA, USA) according to manufacturer’s instructions. RNA concentration and purity were determined by measuring OD260 and OD260/280 ratio, respectively, on a Nano Drop spectrophotometer (Nano-Drop Technologies, Inc., Wilmington, NC, USA); the isolated RNA was kept frozen at −80 °C.(b)Revert Aid H Minus Reverse Transcriptase was used for reverse transcription of total RNA (Thermo Fisher Scientific, USA) to produce cDNA. The cDNA was stored at −20 °C until used. Primers were selected from Primer Bank listed in Table 1.

(c)The relative expression of SIRT1 and P62 genes was detected using a cDNA template by Step One Plus real-time PCR system (Applied Biosystem, Foster City, CA, USA).(d)The conditions of thermal cycler: Denaturation step at 95 °C for 10 min, followed by 40 to 45 amplification cycles (DNA denaturation for 15 s at 95 °C, annealing for 30 s at 60 °C then extension for 30 s at 72 °C). At the end of the last cycle, the temperature raised from 63 to 95 °C for melting curve analysis. The Ct values (cycle threshold) for both (target and housekeeping) genes were estimated, and the relative gene expression assessment was performed using the 2^−ΔΔCt^ method [19].

### 2.9. Morphometric Study

Five non-overlapping random fields (magnification: X400; area: 0.071 mm^2^) of each cerebellar hemisphere of each animal were chosen, photographed, and submitted for morphometric study. Quantification of optical color density and the mean area percentage of GFAP immunohistochemical positive astrocytes in DAB-stained sections was performed using Image software (Media Cybernetics). The sections for S100 were analyzed using ImageJ Software to count the immunoreactive cells in each cutting.

### 2.10. Statistical Analysis

Data were tabulated and statistically analyzed to evaluate the difference between the groups as regards the various parameters using the IBM SPSS (version 21) statistical package (SPSS Inc., Chicago, IL, USA). All data were expressed as the means ± standard deviation (SD). Comparisons between groups were assessed using one-way ANOVA, followed by Post Hoc LSD multiple comparison tests. Results were considered statistically significant when *p* value was <0.05.

## 3. Results

### 3.1. The Ameliorative Effect of Empagliflozin on Histological and Immunohistochemical Results 

Histological examination of the control group cerebellar sections stained with H&E showed the normal histological architecture of the cerebellum: an outer gray matter cortex enclosed by a central white matter core. The cerebellar cortex was composed of three successive layers: outer molecular, middle Purkinje, and inner granular. (Figure 1a). At higher magnification (X400), the granular layer was composed of densely packed clumps of cells with small clear non-cellular areas called cerebellar islands (glomeruli) between them. The Purkinje cell layer was formed out of a continuous single row of pear-shaped cells with vesicular nuclei. The superficial molecular layer was formed out of scattered rounded cells with numerous nerve fibers (Figure 1b).

At higher magnification (X1000), Purkinje cells had well-defined rounded vesicular nuclei and prominent nucleoli with slightly basophilic cytoplasm, and long apical dendrites. Bergmann glial cells (astrocytes) were noticed around the Purkinje cells; they have pale nuclei and pale cytoplasm. The granular layer had groups of rounded cells with rounded heterochromatic nuclei (Figure 1c). Histological examination of the cerebellar specimens of the Empagliflozin-treated group showed the same normal histological architecture of the cerebellum as in the control group with no difference in-between (Figure 1d,e). Examination of the cerebellar specimens of the VGB group revealed loss of the normal cerebellar architecture and degenerative changes. It showed disruption in the linear pattern of the Purkinje layer and some Purkinje cells are irregular, and others are shrunken. Some Purkinje cells have fallen off leaving remarkable empty spaces. Dilated congested blood vessels were seen with an area of hemorrhage in the white matter layer. The vacuolations could also be observed in the granular layer and the outer molecular layer giving them a spongiform appearance. The granular layer contained degenerated granule cells with shrunken pyknotic nuclei and vacuolated cytoplasm. Some neurons with shrunken deeply stained nuclei and vacuolated cytoplasm were seen in the molecular layer (Figure 2a–d).

The administration of empagliflozin significantly modified the VGB-induced histopathological abnormalities. In the VGB+ Empagliflozin group sections, the Purkinje layer retained its normal linear organization, and most of the Purkinje cells regained their normal pyriform shape with few shrunken and deeply stained Purkinje cells in between. The granular layer, the molecular layer, and the white matter core appeared more or less as the control group except for a few vacuolations. Shrunken granular cells with deeply stained nuclei and slightly enlarged glomerular spaces were seen. However, these findings were less frequently observed compared to the VGB group (Figure 3a–c).

GFAP-immunostained cerebellar sections of the control group showed scattered GFAP immunoreactive cells with long and thin processes in the different cerebellar cortical layers (Figure 4a). More abundant GFAP-positive cells appeared in all cerebellar layers of the VGB group that were larger with relatively longer and thicker processes as compared with the control group (Figure 4b,c). In the VGB+ Empagliflozin group, there was a positive expression of GFAP, which was apparently increased as compared to that noticed in the control rats, but less than in group II (Figure 4d). Cerebellar sections of the Empagliflozin-treated group showed scattered GFAP immunoreactive cells in the different cerebellar cortical layers with the same features as in the control group (Figure 4e).

Analysis of S100 immunostaining in the control group showed S100-positive cells located between the Purkinje cells. They are presumably Bergmann glial cells (Figure 5a). The VGB group S100 immunostaining demonstrated a more pronounced diffuse S100 immunostaining with a higher number of S100-positive cells compared with the control group. Furthermore, the intensity of expression was found to be higher (Figure 5b). In the VGB+ Empagliflozin group, there was a positive expression of S 100 immunostaining, which was more or less increased as compared to the control group, but the number and the intensity of expression were found to be less than in the VGB group (Figure 5c). Cerebellar sections of the Empagliflozin-treated group showed the same features as in the control group (Figure 5d).

### 3.2. The Ameliorative Effect of Treatment on Morphometric Results

There was a significant increase in GFAP-immunoreactive astrocytes’ color density and the mean area percentage in the VGB group (86.38 ± 9.04, 38.63 ± 3.31), respectively, compared to the control (45.55 ± 7.00, 20.28 ± 3.83), respectively (*p* < 0.05). It decreased significantly to normal levels in the VGB+ Empagliflozin group (63.77 ± 7.92, 26.88 ± 3.86), respectively, compared to the VGB group (*p* < 0.05) (Figure 4e).

There was a significant increase in the number of S100 immuno-stained cells in the GBV group (7.7 ± 1.56) compared to the control group (14.2 ± 1.87) (*p* < 0.05). The number of S100 immuno-stained cells was significantly restored in the VGB + Empagliflozin group (11 ± 1.33) compared to the VGB group (*p* < 0.05) (Figure 5d).

### 3.3. The Antioxidant Effect of Empagliflozin

The antioxidant status was estimated by measuring SOD activity; it was shown that VGB treatment depressed the enzyme activity significantly when compared with the control and Empagliflozin-treated groups; on the other hand, empagliflozin treatment of diseased animals significantly elevated the activity of SOD when compared with the VGB-treated group (Table 2). A reciprocal effect was observed with the MDA level, VGB elevated its level significantly, while empagliflozin treatment showed correction of its elevated levels significantly as compared to the VGB-treated group. There was a non-significance difference between the control and Empagliflozin-treated groups (Table 2).

### 3.4. The Ameliorative Effect of Empagliflozin on Autophagy 

The level of beclin1 and the relative mRNA gene expression of P62 were estimated to assess the autophagy efflux within the cell. It was found that VGB treatment inhibited the autophagy machinery as observed with low beclin1 level (Table 2) and depressed P62 gene expression (Figure 6) significantly, as compared with the control and Empagliflozin-treated groups. However, empagliflozin relieved this depression in diseased animals and raised the beclin-1 level and relative mRNA gene expression of P62 significantly when compared with VGB treated group.

### 3.5. The Modifying Effect of Empagliflozin on mTOR, AMPK, and Relative mRNA Gene Expression of SIRT1 Levels 

The study herein estimated the level of mTOR, AMPK, and gene expression of SIRT1 as a main regulatory signaling enzyme of the target pathway (Table 2 and Figure 7). VGB treatment significantly suppressed the gene expression of SIRT1 as compared with the control and Empagliflozin-treated groups, while empagliflozin treatment of the diseased group showed significant elevation in SIRT1 gene expression when compared with the VGB-treated group (Figure 7). It was found that mTOR and AMPK levels have a reciprocal attitude.

While the VGB-treated group has a significantly elevated level of mTOR as compared with the control group, the animals within the same group showed a significantly depressed level of AMPK when compared with the control and Empagliflozin-treated animals. On the other hand, cotreatment of the VGB and empagliflozin groups showed significantly depressed levels of mTOR and high levels of AMPK when compared with the VGB treated group.

### 3.6. The Motor Function Improvement as Tested by Behavioral Tests 

VGB treatment induced a marked motor impairment as manifested by a significant reduction in the number of squares crossed by the rats, in the open field activity test, signifying reduced locomotor activity in addition to a significant decrease in the latency time to fall in the rotarod test versus both the control and Empagliflozin-treated groups, designating disturbed motor coordination and balance (*p* < 0.05) (Figure 8). However, cotreatment with empagliflozin significantly alleviated this effect by increasing the number of squares crossed by the rats in the open field activity test, as well as a significant increase in the latency time to fall in the rotarod test (*p* < 0.05) when compared with the VGB group.

## 4. Discussion

It is well acknowledged that mTOR/AMPK signaling dysregulation is a hallmark of numerous neurodevelopmental, neuropsychiatric, and neurodegenerative disorders. Despite VGB’s unique mechanism of action that directly elevates cerebral GABA, its use may be compromised by the development of ocular and cerebellar toxicity; the origins of which remain largely undefined. However, recent studies have clarified that an aberrant GABA-induced mTOR dysregulation with subsequent impaired autophagy could be the potential basis of the VGB-induced retinal and neurotoxicity [7,20].

Plausibly, the activated mTOR pathway elevates the levels of GABA, which, in turn, inhibits mitophagy and pexophagy in yeast cultures. Furthermore, rapamycin, the mTOR inhibitor, abolished GABA-induced defects in yeast, and mammalian cells, as well as in murine models of heritable succinic semialdehyde dehydrogenase deficiency, which is an infrequent GABA metabolism defect. In the latter disorder, by using rapamycin, both mitochondrial numbers and aberrant antioxidant levels that were induced by elevated GABA were also normalized [7]. Moreover, activation of mTOR via phosphorylation, in liver tissue of VGB-treated mice, suggested a strong possibility of similar GABA-induced mTOR activation in ocular and brain tissues [21]. These findings may explain the underlying mechanism of VGB-induced cerebellar toxicity.

An evident mTOR-AMPK signaling pathway has been consistently reported; while both mTOR and AMPK appear to control many pathways reciprocally, there are key lines of crosstalk where both join to regulate many biological processes such as cell metabolism and autophagy. However, AMPK impairs mTORC1 directly, via Raptor phosphorylation and indirectly by the phosphorylating and stimulation of tuberous sclerosis complex 2 (TSC2). Reciprocally, mTORC1 signaling regulates AMPK as S6 kinase phosphorylate and inhibits AMPK [22].

SIRT1 and AMPK are considered to be low-energy sensors that are employed when cells undergo starvation as a part of a transcriptional program that facilitates adaptation to low-nutrient conditions. SIRT1 responds to NAD level alteration and functions as a redox regulator, whereas AMPK recognizes the balance between AMP and ATP in the cytosol. These enzymes arbitrate and strengthen each other’s actions; SIRT1 dampers OS by enhancing antioxidant activity, lessening the inflammatory response to oxygen free radicals, and decreasing the OS lethality. AMPK conserves mitochondrial function, thereby decreasing reactive oxygen species (ROS) formation and it attenuates the resultant proinflammatory and proapoptotic responses [23].

Moreover, both SIRT1 and AMPK are the main regulators of numerous genes that play a target role in maintaining cellular homeostasis, as they can augment autophagy; phosphorylation of AMPK leads to dissociation of the beclin1–Bcl2 complex and promotes maturation of autophagosomes and stimulates their fusion with lysosomes [24], while SIRT1 acts selectively to enhance autophagy [25]. They act directly to maintain the homeostasis of the mitochondrial network, restore peroxisomal function, and improve the antioxidant capacity. Additionally, they interact with the key subunit of NFκB and inhibit its actions, thereby lessening NLRP3 inflammasome activation and muting inflammation-mediated cellular injury [26].

There is undeniable evidence that SGLT2 inhibitors have neuro-protective, cardio-protective, and reno-protective outcomes that are much higher than anticipated, depending on their effects on blood or urinary glucose levels. These achievements cannot be justified by the hypoglycemic effects of SGLT2 inhibitors, as a similar effect is not observed with antidiabetic drugs, which exert a larger antihyperglycemic action [27,28]

In recent reports, SGLT2 inhibitors have shown notable therapeutic benefits through prompting a fasting state, i.e., they induce a transcriptional effect that closely resembles the cell response to starvation. In fact, these drugs activate SIRT1/AMPK and control Akt/mTOR signaling with subsequent autophagy activation, independent of their action on insulin and glucose [29]. Significantly, the effect of SGLT2 to activate the low-energy sensors is not mediated by SGLT2 protein interference, since it is seen in organs that do not have SGLT2 [30].

Indeed, empagliflozin reduced the expression of mTOR complex and Raptor proteins, whereas it increases the expression of ULK1 protein in human renal proximal tubular cells along with improvements in mitochondrial biogenesis, balanced fusion–fission protein expression, reduced mitochondrial ROS production, and inhibiting apoptotic protein expression [31].

A recent study by **Xu et al.** described that empagliflozin administration for 16 weeks in HFD-induced obese mice increased AMPK phosphorylation. It is well-known that activated AMPK stimulates the autophagy machinery throughout numerous mechanisms, including the negative regulation of mTORC1 [32]. Furthermore, it markedly reduced mTOR mRNA expression along with increased phosphorylation at the AMPK N-terminus subunit (Thr172) in obese mice liver tissue; the latter is required for full activation of AMPK. Additionally, empagliflozin activated all steps of the autophagy process as it increased beclin-1 protein levels promoting the nucleation step.

Additionally, it upregulates LC3B expression, which is essential for autophagosome formation. Added to the decreased P62/SQSTM1 protein levels, representing the initiation of autophagic flux, all these effects concluded into increasing the Bcl2/Bax ratio, repressing caspase-8 cleavage, decreasing apoptosis, and conserving cellular integrity [12]. It is noteworthy that autophagy activation can also impair apoptosis via eradication of the p62/SQSTM1 system, which was designed to be lower in the empagliflozin-treated rats. Of note, SQSTM1/p62 is not only an autophagy-specific substrate but also, when accumulated, triggers ROS production and stimulates the DNA damage process [33].

Analysis of molecular docking and invitro recent study estimated that empagliflozin binds glucose transporters in cardiac cells and diminishes glycolysis, reactivates AMPK, and inhibits the mTORC1 pathway activation in a non-diabetic mice model with a nonsignificant effect on fasting blood glucose, suggesting that the beneficial actions on cardiac function were not mediated by empagliflozin’s antihyperglycemic effect [34].

Importantly, empagliflozin did not over-activate AMPK or inhibit mTORC1 in sham groups, rather it restored phosphorylation of AMPK and mTORC1 back to almost normal levels in experimentally induced heart failure [34]. Fukushima, et al. demonstrated that empagliflozin increased the expression of mTOR phosphorylated on serine 2448 mTOR pS2448, which is predominantly contained in mTORC1 and one of the indexes of mTOR activation in obese mice, concomitant with a reduction in p62 accumulations and increased autophagosome formation in autophagosomes in renal proximal cells of obese mice, indicating improvement in the autophagy process upon empagliflozin therapy [35].

Regarding sirtuin 1 (Sirt 1), which is the master regulator of the cellular antioxidant defense mechanism, and matching with our results, Mohamed et al. demonstrated that empagliflozin effectively produced upregulation of renal Sirt 1 expression along with lower renal tissue TNF-α [36]. Moreover, empagliflozin treatment increased hepatic SIRT1/AMPK expression and SIRT1 activity exerting an evident hepatoprotective effect [37].

Concerning neurobehavioral outcomes and in agreement with our results, **Amin et al.** reported that empagliflozin remarkably ameliorated the impaired motor activity and neurological dysfunction as well as histopathological changes in brain tissues of hyperglycemic rats subjected to cerebral I/R injury along with diminished cerebral infarct volume with the lessening of cerebral OS, apoptotic, and inflammatory biomarkers [38].

The possible anti-inflammatory mechanism of empagliflozin is recently explained by Abdelhamid et al., who revealed that empagliflozin efficiently interfered with the NF-κB/Nrf-2/PPAR-γ signaling pathway in an experimental model of alcoholic liver disease [39]. In addition, **Sa-nguanmoo et al.** showed that the anti-inflammatory anti-apoptotic effect of SGLT 2 inhibition exerted a prominent neuroprotective and cognitive-enhancing effect in HFD-induced obese rats [27].

Contradictory to our results, **Youssef et al**. established that empagliflozin administration did not significantly reduce colonic mTOR expression or inflammatory marker levels, in an experimental ulcerative colitis model [40], but we could argue that this can be attributed to the different regimen applied in their study, as the drug was administered for only one week.

In our current study, the detected cerebellar neuropathological toxicity of VGB was in harmony with previous reports [4,41] that used three different doses of VGB. The loss of neuronal cells was detected in the cerebellar cortex of animals with a significant drop in cell counts. There was an increase in the cell loss concomitant with an increase in the VGB dose. **Abd El****-kader et al**. examined the neuropathological findings and performed biochemical assays in the cerebellum of rats after giving oral VGB in two different doses for 4 weeks. They found loss of the normal cerebellar architecture with degenerative changes mainly in the high-dose group. They also reported disturbed liner arrangement of the white matter core fiber, a decrease in the thickness of the granular layer, and a decreased number of Purkinje cells [42].

In this research, the administration of the empagliflozin significantly improved the VGB-induced histopathological changes. GFAP-immunoreactive astrocyte color density and the mean area percentage decreased significantly to normal levels in the VGB+ Empagliflozin group compared to the VGB group. The number of S100 immuno-stained cells was significantly restored in the VGB + Empagliflozin group compared to the VGB group. These results were in accordance with Hayden et al. [43], who proved the protective effect of empagliflozin against glucotoxicity induced in the brain of diabetic mice. Another study detected the ameliorating effect of empagliflozin on the ischemic brain by decreasing neuronal degeneration and exhibiting better rectifying of histopathological changes in the ischemic cerebral cortex and hippocampus [38]. Neurons, microglia, astrocytes, pericytes, and the neurovascular unit mediate homeostasis by regulating the transport between blood and the neural environment. Systemic inflammation increases the permeability of the blood–brain barrier by impairing the endothelium of brain microvessels. In addition, it changes the phenotype of astrocytes and microglia into pro-inflammatory ones with circulating proinflammatory cytokines [44]. In a mouse model of T2DM, empagliflozin had a protective effect, involving a remodeling effect on the blood–brain barrier, astrocytes, microglia, pericytes, oligodendrocytes, and the neurovascular unit that is composed of endothelial cells lining brain microvessels. In addition, empagliflozin significantly increases leukocyte expression of antioxidative enzymes including glutathione s-reductase and catalase [45].

## 5. Conclusions

Empagliflozin efficiently ameliorated the VGB-induced disrupted mTOR/AMPK/SIRT-1 signaling axis with subsequent improvement in the autophagy process and mitigation of the oxidative and inflammatory cellular environment. Such an innovative therapeutic approach offers a preclinical basis for future use of empagliflozin as an adjuvant for alleviation of vigabatrin-associated cerebellar toxicity.

## 6. Limitations

To extrapolate our findings, and to have a better deduction mechanism, in vitro experiments should be performed as a subsequent step to support data escalation and pave the way for further clinical applications. Likewise, the use of ELISA, without confirmation of the protein target by Western blotting, is considered to be one of the study’s limitations. This also leads to a future assessment of the interplay between the chief proteins identified in this study.

## Figures and Tables

**Figure 1 molecules-27-03659-f001:**
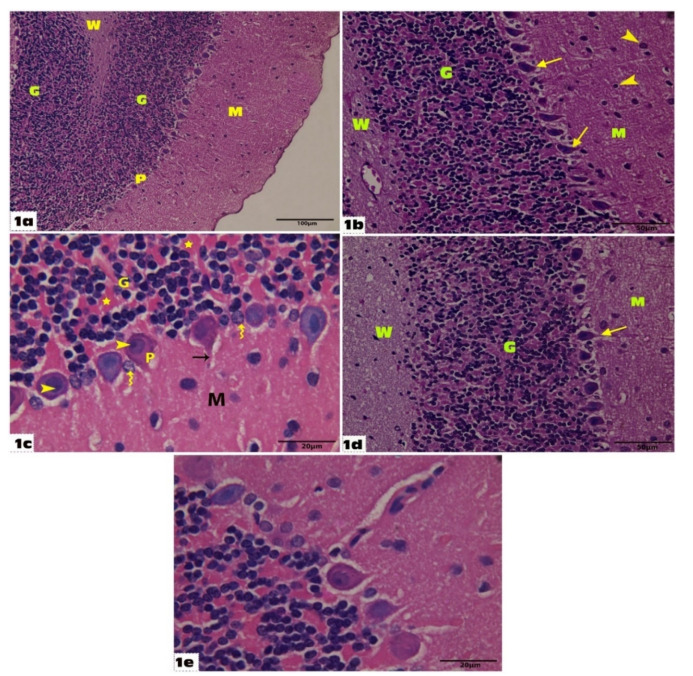
Sections of the cerebellum of the control and Empagliflozin-treated groups (**group I and IV**): (**a**) Shows the control cerebellar cortex lamination into outer molecular (M), middle Purkinje layer (P), and inner granular (G), and inner white matter of cerebellar medulla (W) (H&E 200). (**b**) Shows the molecular layer (M) has scattered rounded cells (arrow heads) with numerous nerve fibers. Purkinje cells (arrow) arranged in a single row of large pear-shaped cells with vesicular nuclei. The granular layer (G) shows densely packed clumps of cells with small clear non-cellular areas called cerebellar islands (glomeruli) in the control group (H&E 400). (**c**) Shows flask-shaped Purkinje cells (P), which contain well-defined rounded vesicular nuclei and prominent nucleoli (arrow heads) with slightly basophilic cytoplasm, and long apical dendrites (arrows). Bergmann glial cells (astrocytes) are noticed around the Purkinje cells; they have pale nuclei and pale cytoplasm (wavy arrows). The granular layer has groups of rounded cells with rounded heterochromatic nuclei and is separated by small clear non cellular areas called cerebellar islands (glomeruli) (Stars) in the control group (H&E 1000). (**d**) Shows histological architecture of the cerebellum in the Empagliflozin-treated group with the same features as in the control group. Purkinje-cells (arrow) arranged in a single row of large pear-shaped cells with vesicular nuclei and the granular layer (G) shows densely packed clumps of cells (H&E 400). (**e**) Shows histological architecture of the cerebellum in the Empagliflozin-treated group with the same features as in the control group (H&E 1000).

**Figure 2 molecules-27-03659-f002:**
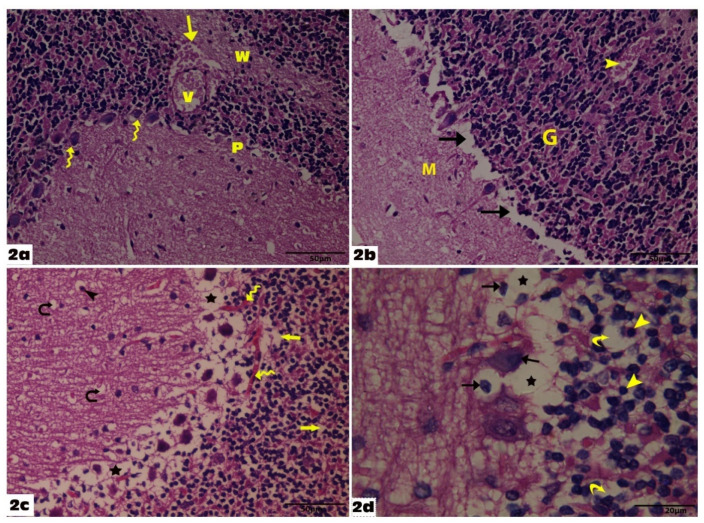
Sections from the rat cerebellum of group II: (**a**): Show disruption in the linear pattern of the Purkinje layer and some Purkinje cells are irregular, and others are shrunken (wavy arrows). Dilated congested blood vessel (V) is seen with area of hemorrhage (arrow) in the white matter layer (H&E 400). (**b**): Shows marked disarrangement of the Purkinje layer, some Purkinje cells have fallen off leaving remarkable empty spaces (Arrows), and congested blood vessels (arrowhead) (H&E 400). (**c**): Shows obvious degenerative changes especially in the Purkinje (P) layer, in the form of marked vacuolated empty spaces (stars), and the Purkinje cells are deformed. There are increased and congested blood vessels in the Purkinje layer (wavy arrows), which also shows empty spaces scattered in the granular (Arrows) and in the molecular layer (curved arrows), and some neurons with shrunken deeply stained nuclei and vacuolated cytoplasm (Arrowhead) are seen in the molecular layer (H&E 400). (**d**): Shows the granular layer with many deformed cells with shrunken pyknotic nuclei and vacuolated cytoplasm (Arrowheads); other cells have degenerated leaving empty spaces (curved arrows). The Purkinje layer showed deformed Purkinje cells with deeply stained pyknotic nuclei (arrows) and marked cytoplasmic vacuolations (stars) (H&E 1000).

**Figure 3 molecules-27-03659-f003:**
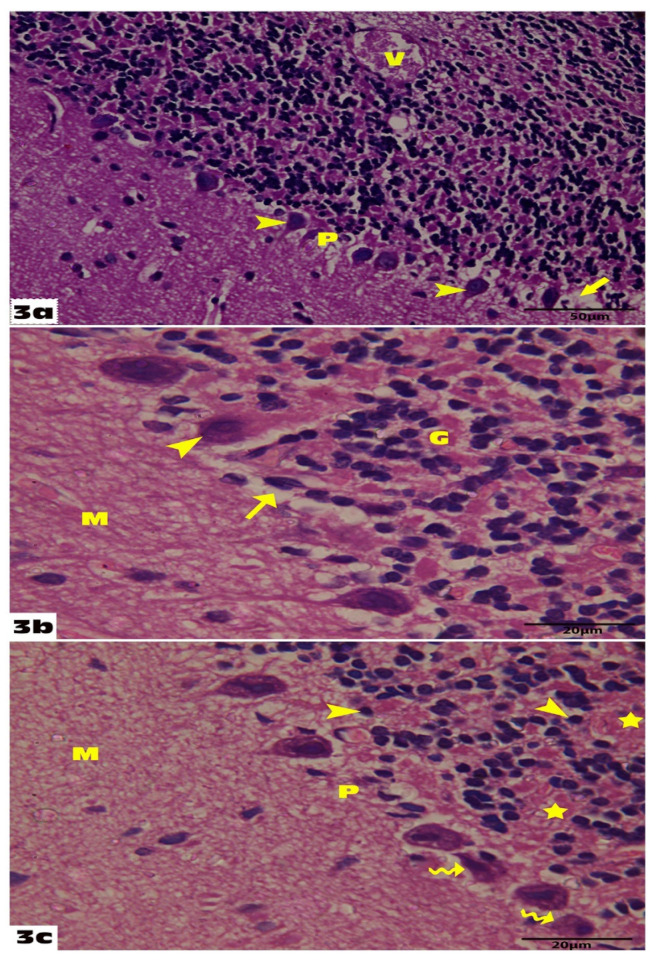
Sections from the rat cerebellum of group III: (**a**): Show that the Purkinje layer (P) retains its normal linear organization showing some normal cells (arrowheads), but other cells show irregular form and vacuolar spaces (arrow), and the granular layer shows nearly normal clumps of cells and shows a congested blood vessel (V) (H&E 400). (**b**): Shows irregular Purkinje cell with deeply stained pyknotic nucleus (arrowhead) and small vacuolated empty spaces (arrow); the remaining Purkinje cells are normal. The molecular layer (M) with scattered neurons appears nearly normal (H&E 1000). (**c**): Shows the granular layer with few shrunken cells with deeply stained nuclei (arrowhead), slightly enlarged glomerular spaces (stars), and the Purkinje layer shows normal organization with few irregular Purkinje cells (wavy arrows) (H&E 1000).

**Figure 4 molecules-27-03659-f004:**
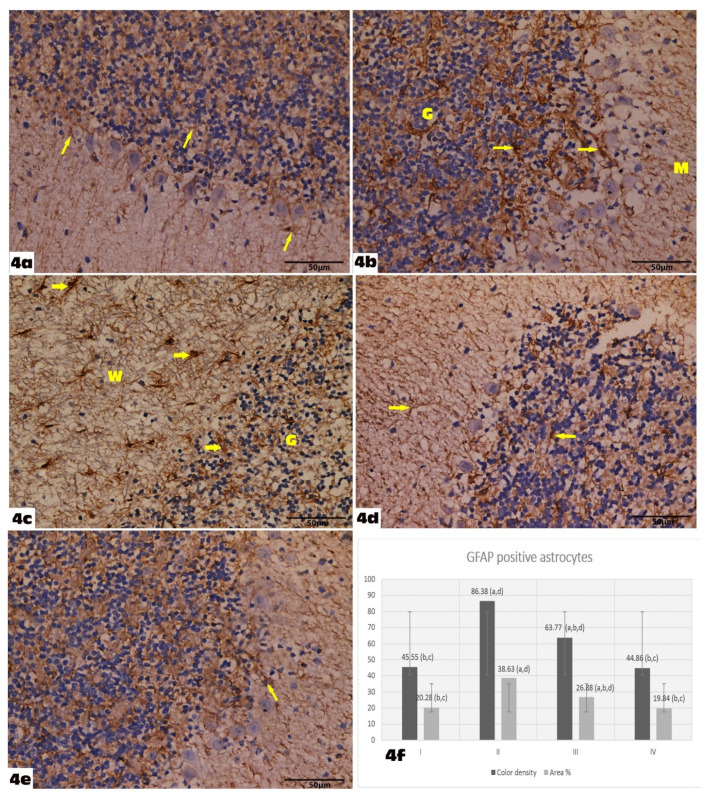
Shows Glial fibrillary acidic protein (GFAP) immunohistochemically stained cerebellar sections: (GFAP immunostain X 400). (**a**): Control group showing scattered GFAP immunoreactive cells with long and thin processes (arrows) in the different cerebellar cortical layers. (**b**,**c**): Group II showing more abundant GFAP-positive cells in all cerebellar layers molecular (M), Purkinje (P), Granular (G), and in the medullary white matter (M). These cells appear larger with relatively longer and thicker processes (arrows) as compared with the control group. (**d**): Group III showing positive expression of GFAP, which is apparently increased as compared to that noticed in the control rats, but lesser than group II (arrows). (**e**): Group IV showing scattered GFAP immunoreactive cells (arrow) in the different cerebellar cortical layers with the same features as in the control group. (**f**): Graphic representation of the morphometric results of glial fibrillary acidic protein optical density. ^a^ denotes a statistically significant difference compared to group I; ^b^ denotes a statistically significant difference compared to group II; ^c^ denotes a statistically significant difference compared to group III; ^d^ denotes a statistically significant difference compared to group IV, by one-way analysis of variance followed by post hoc Tukey’s test.

**Figure 5 molecules-27-03659-f005:**
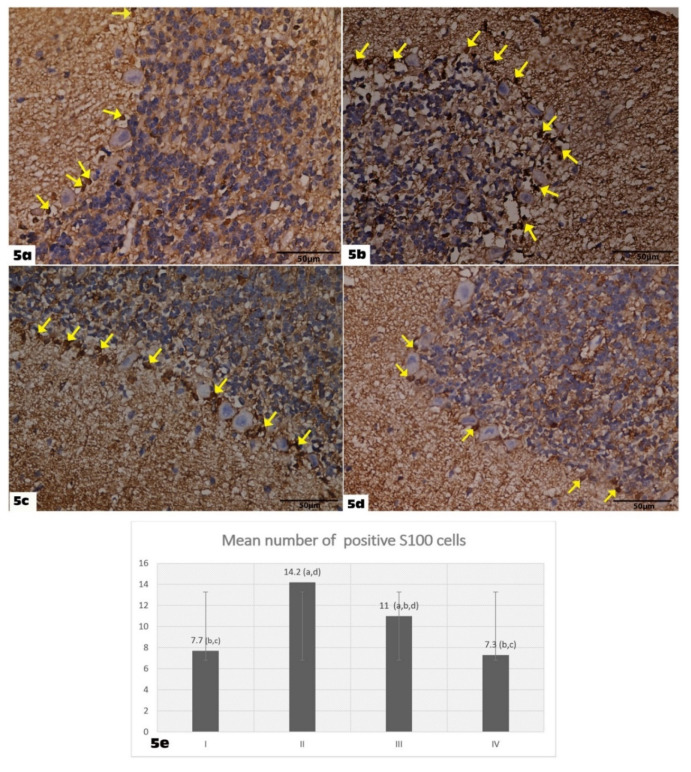
Shows S100 immunohistochemically stained cerebellar sections: (S100 immunostain X400) (**a**): Control group showing S100-positive cells located between the Purkinje cells (yellow arrows). (**b**): Group II showing more pronounced diffuse S100 immunostaining with a higher number of S100-positive cells (yellow arrows). (**c**): Group III showing positive expression of S100 immunostaining with S100-positive cells are lesser in number than the VGB group (yellow arrows). (**d**): Group IV showing S100-positive cells located between the Purkinje cells (yellow arrows). (**e**): Graphic representation of the morphometric results of a mean number of positive S100 immuno-stained cells. ^a^ denotes a statistically significant difference compared to group I; ^b^ denotes a statistically significant difference compared to group II; ^c^ denotes a statistically significant difference compared to group III; ^d^ denotes a statistically significant difference compared to group IV, by one-way analysis of variance followed by post hoc Tukey’s test.

**Figure 6 molecules-27-03659-f006:**
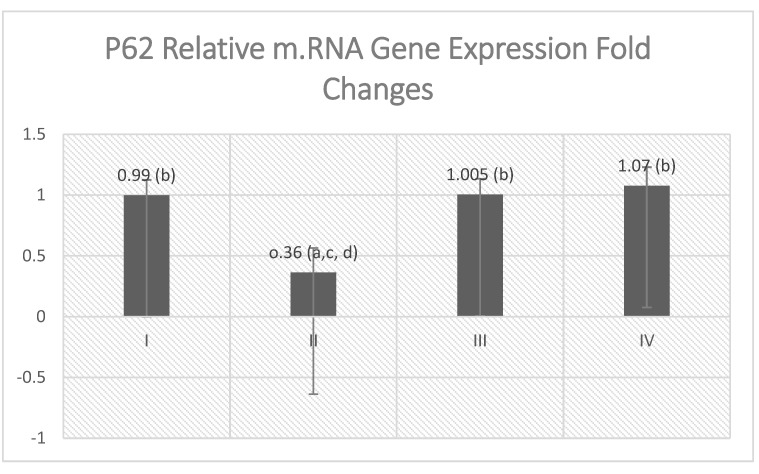
Effect of empagliflozin treatment on P62gene expression. Values are represented as mean ± SD (n = 10). ^a^ denotes a statistically significant difference compared to group I; ^b^ denotes a statistically significant difference compared to group II; ^c^ denotes a statistically significant difference compared to group III; ^d^ denotes a statistically significant difference compared to group IV, by one-way analysis of variance followed by post hoc Tukey’s test.

**Figure 7 molecules-27-03659-f007:**
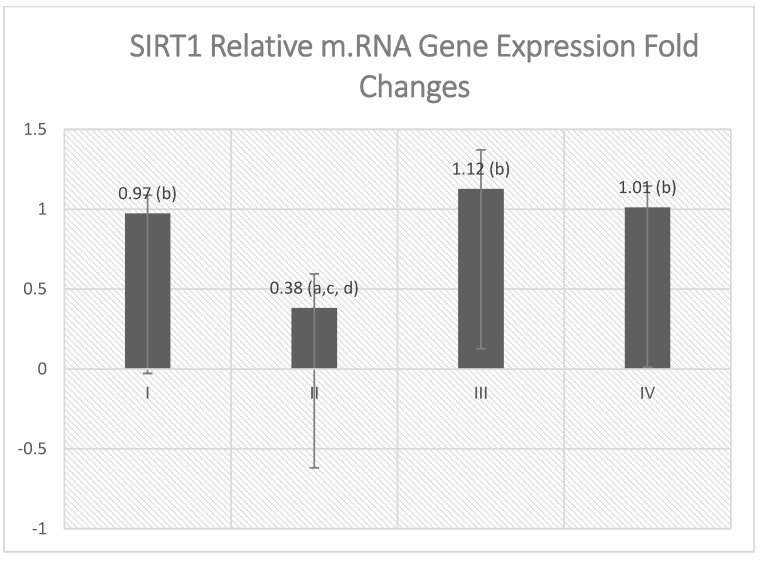
Effect of empagliflozin treatment on SIRT1 gene expression. Values are represented as mean ± SD (n = 10). ^a^ denotes a statistically significant difference compared to group I; ^b^ denotes a statistically significant difference compared to group II; ^c^ denotes a statistically significant difference compared to group III; ^d^ denotes a statistically significant difference compared to group IV, by one-way analysis of variance followed by post hoc Tukey’s test.

**Figure 8 molecules-27-03659-f008:**
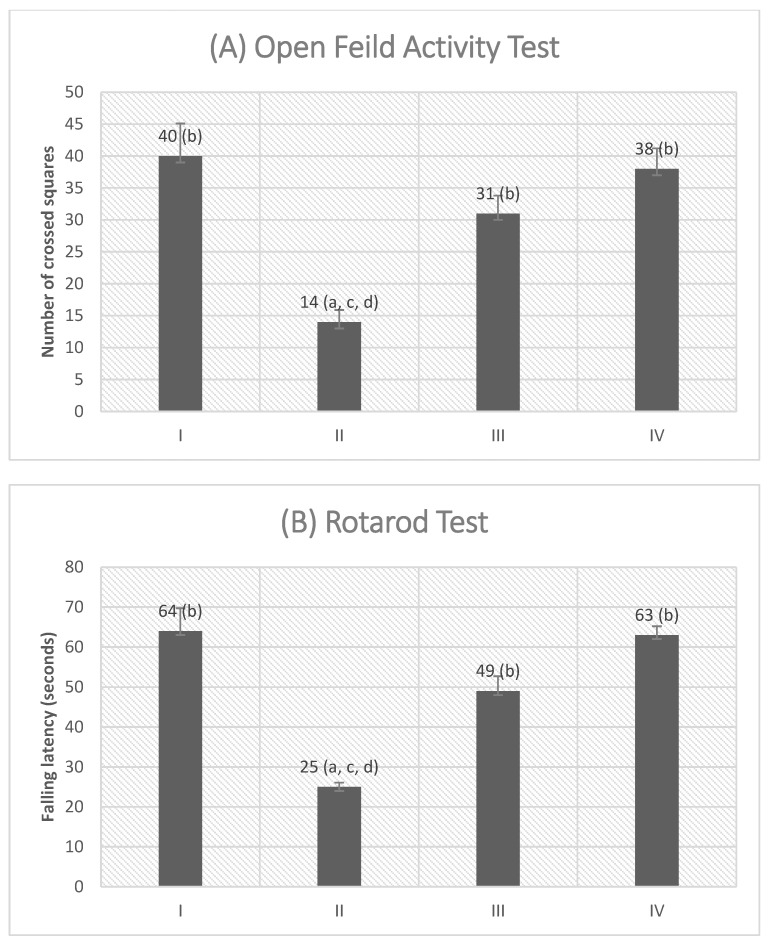
Effects of empagliflozin treatment on the motor behavioral deficit of vigabatrin-treated rats. Locomotor activity (**A**) is presented as the number of crossed squares in the open field arena. Motor coordination (**B**) is presented as falling latency on the rotarod apparatus (seconds). Values are represented as mean ± SD (n = 10). ^a^ denotes a statistically significant difference compared to group I; ^b^ denotes a statistically significant difference compared to group II; ^c^ denotes a statistically significant difference compared to group III; ^d^ denotes a statistically significant difference compared to group IV, by one-way analysis of variance followed by post hoc Tukey’s test.

**Table 1 molecules-27-03659-t001:** Primers of SIRT1 and P62.

Primer for SIRT1:	
** *Forward:* **	5′-CCAGCCATCTCTCTGTCACA-3′
** *Reverse:* **	5′-TGGTTTCATGATAGCAAGCGG-3′
**Primer for P62:**	
** *Forward:* **	5′- GCACCCCAATGTGATCTG C -3′
** *Reverse:* **	5′- CGCTACACAAGTCGTAGTCTGG -3′
**Primer Housekeeping gene GAPDH:**	
** *Forward:* **	5′-CCACTCCTCCACCTTTGAC-3′
** *Reverse:* **	5′-ACCCTGTTGCTGTAGCCA-3′

**Table 2 molecules-27-03659-t002:** Effect of Empagliflozin treatment on m.TOR, AMPK, Redox status, and autophagy among the studied groups.

*Parameter*	Control (I) ^a^ N = 10	Vigabtrin (II) ^b^ N = 10	Vigabtrin + Empagliflozin (III) ^c^ N = 10	Empagliflozin (IV) ^d^ N = 10
*SOD (Activity units/mg protein)*	429.312 ± 10.8 ^b^	357.2 ± 22.45 ^a,c,d^	437.8 ± 11.23 ^b^	431.012 ± 12.2 ^b^
*MDA (nmol/mg protein/mL)*	58.46 ± 5.2 ^b^	152.608 ± 6.4 ^a,c,d^	60.36 ± 9.6 ^b^	56.98 ± 4.2 ^b^
*m.Tor (pg/mL)*	9.25 ± 1.2 ^b^	21.22 ± 2 ^a,c,d^	8.458 ± 9 ^b^	8.98 ± 2.1 ^b^
*AMPK (ng/mg protein/mL)*	1.74 ± 0.4 ^b^	0.95 ± 0.2 ^a,c,d^	1.62 ± 0.3 ^b^	1.79 ± o.3 ^b^
*Beclin1 (ng/mg protein/mL)*	1.096 ± 0.16 ^b^	0.314 ± 0.17 ^a, c, d^	1.268 ± 0.22 ^b^	1.13 ± 0.23 ^b^

^a, b & c^ denote a statistically significant difference at (*p* < 0.05) using one-way ANOVA with Tukey’s post hoc test. ^a^ Denotes statistical significance when compared to group I. ^b^ Denotes statistical significance when compared to group II. ^c^ Denotes statistical significance when compared to group III. ^d^ Denotes statistical significance when compared to group IV.

## Data Availability

Data sharing is applicable to this article as new data were analyzed in this study. All data that support the finding of the current study are available from R.A.E.G., upon reasonable request.

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
