# Peer review of "The Ameliorative Effect of Empagliflozin in Vigabatrin-Induced Cerebellar/Neurobehavioral Deficits: Targeting mTOR/AMPK/SIRT-1 Signaling Pathways"

_molecules, 2022, doi:10.3390/molecules27123659_

Round 1

Reviewer 1 Report

with all due respect, none of the previous comments were answered which are major flaws in the whole study design and experimentation. the authors did not take any step to rectify these comments. Thus, the work can not be accepted in the current form and should be rejected. Unfortunately, the current work harbors the same weaknesses that are not answered nor being rectified from the previous two submissions. 

the previous comments related to the cohort number justification along with the Western blotting analysis pf the p-Tau

Thus the paper is based without proper statistics 

the need for the cohort of empagliflozin is highly needed

the inclusion of NeuN is absent

Author Response

-The previous comments related to the cohort number justification along with the Western blotting analysis pf the p-Tau

Replay

with respect to your valuable comment, we have limited fund and time, hence we added this point in the limitation.

- The need for the cohort of empagliflozin is highly needed

Replay

with respect to your valuable comment, we have already added this group.

- The inclusion of NeuN is absent

replay

  • With all respect to your valuable notion, in our research we assessed neuronal injury by h&e and biological evaluation. We used GFAP for astrocytes evaluation and S100 for bergmann cells evaluation which are glial cells responsible for regeneration.

Reviewer 2 Report

In the abstract in the method section the Group IV is missing

How did you choose the GAPDH as your housekeeping gene? Usually three housekeeping gene is used. If you didn’t do the estimation of the most suitable reference genes by GeNorm please omit the RT-PCR data from the manuscript. Because the end results can have significant differences in the analysis of a target gene expression if the wright reference/hosekeeping gene is not used.

In the results:

  1. In Figure 1. add the picture with higher magnification of the Group IV so it could be compared to the same picture in Grope I
  2. In Fig 4. Ad the picture of GFAP staining of Group IV
  3. In Fig 5. again, ad the staining of group IV and in this picture, I do not see the increase of S100 positive cell in Purkinjean layer. How did you count the cells because the number seems to be the same in Group II and III?
  4. In the section 3.2 line 370 you are missing the letter V

Author Response

In the abstract in the method section the Group IV is missing
Replay
We have added this group in the abstract.
How did you choose the GAPDH as your housekeeping gene? Usually three housekeeping gene is used. If you didn’t do the estimation of the most suitable reference genes by GeNorm please omit the RT-PCR data from the manuscript. Because the end results can have significant differences in the analysis of a target gene expression if the wright reference/hosekeeping gene is not used.
Replay
First
for P62 mRNA gene expression we have chosen GAPDH as our housekeeping gene according to the previous works of:
- (R. Gómez-Sánchez et al. / Data in Brief 7 (2016) 641–647: mRNA and protein dataset of autophagy markers (LC3 and p62) in several cell lines)
- (L.-z. Zhang et al. / Saudi Journal of Biological Sciences 26 (2019) 1986–1990: Correlation between PTEN and P62 gene expression in rat colorectal cancer cel)
Second
for SIRT1 mRNA gene expression we have chosen GAPDH as our housekeeping gene according to the previous works of:
- (Chen , et. Al. High levels of SIRT1 expression enhance tumorigenesis and associate with a poor prognosis of colorectal carcinoma patients: SCIENTIFIC REPORTS | 4 : 7481 | DOI: 10.1038/srep07481)
- (Barbagallo et al. Silibinin Regulates Lipid Metabolism and Differentiation in Functional Human Adipocytes, Frontiers in Pharmacology · January 2016)

 In Figure 1. add the picture with higher magnification of the Group IV so it could be compared to the same picture in Grope I
Replay
We have added it to assure our result. But, we see that there is no need to add it as it is the same as the control group.
 In Fig 4. Ad the picture of GFAP staining of Group IV
Replay
We have added it to assure our result. But, we see that there is no need to add it as it is the same as the control group.

  In Fig 5. again, ad the staining of group IV and in this picture, I do not see the increase of S100 positive cell in Purkinjean layer. How did you count the cells because the number seems to be the same in Group II and III?
Replay
-We have added it to assure our result. But, we see that there is no need to add it as it is the same as the control group.
-S100 for bergmann cells evaluation which are glial cells responsible for regeneration and present inbetween purkinje cells as showing in the figure (5) by arrows.
-The results S100 positive cells in Group II and III is correct. For reassurance, this is the spss output for S 100 resullts.
  •  
Report
s100number 
groups
Mean
N
Std. Deviation
1
7.70
10
1.567
2
14.20
10
1.874
3
11.00
10
1.333
Total
10.97
30
3.113

Multiple Comparisons
Dependent Variable:   s100number 
LSD 
(I) groups
(J) groups
Mean Difference (I-J)
Std. Error
Sig.
95% Confidence Interval
Lower Bound
Upper Bound
1
2
-6.500-*
.719
.000
-7.97-
-5.03-
3
-3.300-*
.719
.000
-4.77-
-1.83-
2
1
6.500*
.719
.000
5.03
7.97
3
3.200*
.719
.000
1.73
4.67
3
1
3.300*
.719
.000
1.83
4.77
2
-3.200-*
.719
.000
-4.67-
-1.73-
*. The mean difference is significant at the 0.05 level.

 In the section 3.2 line 370 you are missing the letter V
Replay
We have corrected it.

Reviewer 3 Report

The author showed the beneficial effect of Empagliflozin on inhibiting astrocytes activation and oxidative stress as well as promoting autophagic pathway and neurological behaviors deficits in the long-term Vigabatrin-treated rats. The study is well-designed and the conlusion is solid. here are some comments.

  1. Line21 Group IV is missing
  2. S100 and GFAP are both astrocytes markers. It is well known that astrocytes activation involving in the inflammation, oxidative stress and BBB disruption. Discussion about the beneficial effects of Empagliflozin on inhibiting astrocytes activation is needed.
  3. Line126 “starting on day 1 with the first dose of vigabatrin for 6 consecutive weeks [13]”this group is only empagliflozin treated group without vigabatrin.

Author Response

 Line21 Group IV is missing
Replay
We have added it.
S100 and GFAP are both astrocytes markers. It is well known that astrocytes activation involving in the inflammation, oxidative stress and BBB disruption. Discussion about the beneficial effects of Empagliflozin on inhibiting astrocytes activation is needed.
Replay
We have added it.
Line126 “starting on day 1 with the first dose of vigabatrin for 6 consecutive weeks [13]” this group is only empagliflozin treated group without vigabatrin.
Replay
We have corrected it.

Round 2

Reviewer 1 Report

the whole work is substandard and if the authors can not afford doing extra work, they should not submit it here.

The quantification of NeuN as well as having a statistical justification for the cohort numbers

he absence of an extra animal cohort is needed

quantification of NeuN is required 

justification of animals and power analysis are lacking 

Author Response

The absence of an extra animal cohort is needed

Replay

with respect to your valuable comment, we have already added this group.

quantification of NeuN is required 

Replay

  • With all respect to your valuable notion, in our research we assessed neuronal injury by h&e and biological evaluation. We used GFAP for astrocytes evaluation and S100 for bergmann cells evaluation which are glial cells responsible for regeneration.
  • justification of animals and power analysis are lacking 
  • Replay
  • with respect to your valuable comment, All data was tested by the Kolmogorov-Smirnov Goodness of Fit Test (K-S test) as a test for normality and data was normally distributed.

This manuscript is a resubmission of an earlier submission. The following is a list of the peer review reports and author responses from that submission.

Round 1

Reviewer 1 Report

we thank the reviewers for the comments, the manuscript was already reviewed and there were major deficits and experimental flaws that were highlighted and there were requests for additional experiments.

Unfortunately, the current work harbors the same weaknesses that are not answered nor being rectified.

1- The power analysis or justification  for using n=10 needs to be calculated as power analysis and the selection of n=9 or 12 or 10 or 15 is not arbitrary and needs to be calculated which is obviously not done in the experiments.

2- the request of performing western blotting technique is requested to look at phosphorylation levels and depicts activation status, which is usually not quantitated via ELISA

3- thee is a need to assess  NeuN to assess neuronal injury and iba-1 to assess microglial activation. S100 as described by the authors does not represent a marker of glial protein but is expressed in Nuclear and cytoplasmic expression mainly in CNS, peripheral nerves, adipocytes and subset of lymphoid cells.

Finally, the need for the cohort of empagliflozin is needed as a cohort to highlight the adverse effect of this drug. after all you are performing a clinical study where I am asking the prescription of this drug to patients, this is an experimental study requiring this important group.

   The above comments represent major flaws in the presented work

Author Response

The power analysis or justification for using n=10 needs to be calculated as power analysis and the selection of n=9 or 12 or 10 or 15 is not arbitrary and needs to be calculated which is obviously not done in the experiments.

Replay

  • All data was tested by the Kolmogorov-Smirnov Goodness of Fit Test (K-S test) as a test for normality and data was normally distributed.

The request of performing western blotting technique is requested to look at phosphorylation levels and depicts activation status, which is usually not quantitated via ELISA

Reply:

  • With all respect to your valuable notion, in our research we used ELISA technique and not western for two reasons
  • ELISA is a very sensitive and very sophisticated method that detects the presence of antigen and antibody in the tissue/blood, while western blot is a technique that detects a specific protein from a protein mixture. Elisa is qualitative as well as quantitative. In contrast, western blot is qualitative. Sometimes it is semi-quantitative. When considering the time taken for the test, Elisa test is time-consuming while western blot is more time consuming than Elisa.
  • As a group of researchers, we didn’t receive any fund to perform our research, thus it would be of a great financial burden to perform both of them together in addition to the PCR technique and other spectrophotometric parameters.

 There is a need to assess NeuN to assess neuronal injury and iba-1 to assess microglial activation. S100 as described by the authors does not represent a marker of glial protein but is expressed in Nuclear and cytoplasmic expression mainly in CNS, peripheral nerves, adipocytes and subset of lymphoid cells.

Reply

  • With all respect to your valuable notion, in our research we assessed neuronal injury by h&e and biological evaluation. We used GFAP for astrocytes evaluation and S100 for bergmann cells evaluation which are glial cells responsible for regeneration.

Finally, the need for the cohort of empagliflozin is needed as a cohort to highlight the adverse effect of this drug. after all you are performing a clinical study where I am asking the prescription of this drug to patients, this is an experimental study requiring this important group.

Replay

We did not use empagliflozin alone in an experimental group as we thought that it would be clinically irrational to prescribe a drug for healthy subjects, from risk-benefit relationship point of view; empagliflozin is not a risk-free drug, as it is associated with hypotension, ketoacidosis, genital mycotic infections and pyelonephritis in addition to possible drug interactions. (Sizar, O., Podder, V., & Talati, R. (2020). Empagliflozin. In StatPearls [Internet]. StatPearls Publishing.‏)

Reviewer 2 Report

I am satisfied with the revision. Accepted as is.

Author Response

thank you for your valubale comment

Reviewer 3 Report

Here are some suggestions to strengthen the work.

  1. Line101:Vigabatrin was purchased from Sigma‐Aldrich (St. Louis, MO) while Empagliflozin was obtained from Cayman Chemical (MI). catalogue numbers for both drugs are needed.
  2. Line106: what age of rat used ?
  3. Line116: DMSO concentration are needed.
  4. Line116: dose the oral administration here mean oral administration gavage ? what is the volume of drugs ?
  5. Provide rational for the concentration and duration selection of  vigabatrin and empagliflozin.
  6. Four groups are suggested. The empagliflozin only treated group is lack. 
  7. A work flow for experimental design is needed including all the experiments time and order.
  8. What is the tissue used for detecting SOD, MDA, mTOR, Beclin1, SIRT1, P62 ? cerebellum ?
  9.  Line 133, how to record the number of squares crossed ? by video or by eye ? information is needed. Whether the total travelling distance changed or not ?
  10. Subtitle of results section should be the main findings.
  11. For all the histological images, the scale bar is needed.
  12. For all the histological images, the group name for each image and the labelled protein should be clearly shown in the figures, not in the legends.
  13. Figure1-5, all the statistic analysis are needed and should be included in the figures beside the images. Without the bar graph, any conclusion can not be made.
  14. The author need to reorgnized the order of figures. The recent version make people confused.

Author Response

Reviewer 3

Line101: Vigabatrin was purchased from Sigma‐Aldrich (St. Louis, MO) while Empagliflozin was obtained from Cayman Chemical (MI). catalogue numbers for both drugs are needed.

  • Both catalogue numbers have been added.

Line106: what age of rat used?

  • 9 weeks

Line116: DMSO concentration are needed.

  • The concentration has been added.

Line116: dose the oral administration here mean oral administration gavage ? what is the volume of drugs ?

  • Yes, oral administration means drug administration by oral gavage.
  • Regarding vigabatrin, the final concentration was 25mg in each one milliliter of 0.9% sodium chloride
  • Regarding empagliflozin, the final concentration was 2 mg in each one milliliter of DMSO

Provide rational for the concentration and duration selection of vigabatrin and empagliflozin.

  • We chose the concentration and duration of vigabatrin based on previous report by Singhet al., 2013 , Singh and his colleagues studied three different doses of vigabatrin 125 , 250 and 500 mg/kg body weight and observed that 250 and 500 mg/kg vigabatrin induced significant neuronal cell loss in the cerebellar cortex of rats whereas 125 mg/kg body weight did not induce such histological changes.
  • We chose the concentration and duration of empagliflozin based on recent report by Nasiri-Ansari et al., 2021 who proved that empagliflozin treatment 10 mg/kg/day for five weeks can significantly activate autophagy via increased AMPK phosphorylation, decreased mTOR and alleviate cellular apoptosis. These cellular mechanistic pathways are the same druggable targets of our study.
  • References:

Deepa Singh , Sunder Lal Jethani  , Aksh Dubey: Vigabatrin induced Cell loss in the Cerebellar Cortex of Albino Rats. Journal of Clinical and Diagnostic Research. 2013 Nov, Vol-7(11): 2555-2558. DOI: 10.7860/JCDR/2013/6187.3610

  1. Nasiri-Ansari, C. Nikolopoulou, K. Papoutsi, I. Kyrou, C. S. Mantzoros, G. Kyriakopoulos, E. Kassi, Empagliflozin Attenuates Non-Alcoholic Fatty Liver Disease (NAFLD) in High Fat Diet Fed ApoE (-/-) Mice by Activating Autophagy and Reducing ER Stress and Apoptosis, International Journal of Molecular Sciences. 22(2) (2021) 818.‏

Four groups are suggested. The empagliflozin only treated group is lack. 

Reply

  • We did not use empagliflozin alone in an experimental group as we thought that it would be clinically irrational to prescribe a drug for healthy subjects, from risk-benefit relationship point of view; empagliflozin is not a risk-free drug, as it is associated with hypotension, ketoacidosis, genital mycotic infections and pyelonephritis in addition to possible drug interactions. (Sizar, O., Podder, V., & Talati, R. (2020). Empagliflozin. In StatPearls [Internet]. StatPearls Publishing.‏)

A work flow for experimental design is needed including all the experiments time and order.

  • We appreciate your valuable advice, a work flow for experimental design is included in the modified manuscript.

What is the tissue used for detecting SOD, MDA, mTOR, Beclin1, SIRT1, P62? cerebellum?

  • Yes, the above parameters have been detected in cerebellum.

Line 133, how to record the number of squares crossed? by video or by eye? information is needed. Whether the total travelling distance changed or not?

  • The number,of squares crossed with all paws was observed and counted manually by an observer in a 6-minutes session. The observer was blind to the animal condition.
  • Total travelling distance was not recorded in our study.

Subtitle of results section should be the main findings.

  • All subtitles have been changed according to finding

For all the histological images, the scale bar is  needed.

  • The scale bar was added to all histological images.

For all the histological images, the group name for each image and the labelled protein should be clearly shown in the figures, not in the legends.

-With respect to your valuable comment, adding group name and labelled protein on the figures themselves will hide the details of the figures. Also figures legends explain all the details easily.

Figure1-5, all the statistic analysis are needed and should be included in the figures beside the images. Without the bar graph, any conclusion can not be made.

  • Graphic representation of the morphometric results (figures 4&5) in clustered columns was added in the figures beside the images.
  • Figures (1-3) do not include graphs as they demonstrates histological features in   different groups by H&E staining.
  • If the comment needs to change clustered columns to bar graph, we thought that clustered columns for these values and analysis is better With respect to your valuable comment.

The author need to reorganize the order of figures. The recent version make people confused

Replay

  • With respect to your valuable comment, all figure and table numbers are cited in their correct place in the result section which make it easy for the reader to understand.

Reviewer 4 Report

The study describing the effects of Empagliflozin on Vigabatrin-induced neurobehavioral deficits is very innovative and well planned and executed. The combination treatment can be developed in the future for clinical testing. The manuscript is nicely written with appropriate data sets and therefore can be accepted after considering the following minor points.

  • The title is difficult to read. Make it short and clear.
  • How was the dose of Empagliflozin selected for in vivo study? Provide information on Empagliflozin solubility.
  • How behavioral data are recorded? Manually or using video tracking software. Provide representative movement path of different group mice for the open field test.
  • Cite figures in the results section at appropriate places rather than just mentioning them at the end of the paragraph.
  • Present all the data at mean±
  • In figure 10, difficult to believe significant differences between groups for S100 results. Recheck the statistics.
  • Correct grammatical errors and provide abbreviations throughout the manuscript.
  • Provide primary references in the introduction and discussion sections.

Author Response

Reviewer 4

The title is difficult to read. Make it short and clear.

Reply

  • Thank you for this valuable suggestion, we modified the title as (The Ameliorative Effect of Empagliflozin in Vigabatrin-Induced Cerebellar /Neurobehavioral Deficits: Targeting mTOR/AMPK/ SIRT-1 Signaling Pathways).

  • How was the dose of Empagliflozin selected for in vivo study? Provide information on Empagliflozin solubility.

Replay

  • We chose the dose of Empagliflozin based on recent report by Nasiri-Ansari et al., 2021 who proved that empagliflozin treatment 10 mg/kg/day for five weeks can significantly activate autophagy via increased AMPK phosphorylation, decreased mTOR and alleviate cellular apoptosis. These cellular mechanistic pathways are the same druggable targets of our study.
  • Regarding Empagliflozin solubility; Empagliflozin is soluble in Dimethylformamide (30 mg/mL), Dimethylsulfoxide (30 mg/mL), Ethanol (30 mg/mL) and Ethanol: PBS (0.5 mg/mL). Reference: https://www.caymanchem.com/product/17375/empagliflozin.

How behavioral data are recorded? Manually or using video tracking software. Provide representative movement path of different group mice for the open field test.

Replay

  • Behavioral data were recorded manually by an observer who was blind to the animal condition.
  • Unfortunately, we do not have video tracking software, so movement path of different groups was not recorded.

Cite figures in the results section at appropriate places rather than just mentioning them at the end of the paragraph.

  • All figures are cited in their appropriate place.

Present all the data at mean±

  • We already represented in that way as in table (2)

In figure 10, difficult to believe significant differences between groups for S100 results. Recheck the statistics.

  • We rechecked the statistics, the results are correct. For reassurance, this is the spss output for S 100 resullts.
  •  

Report

s100number 

groups

Mean

N

Std. Deviation

1

7.70

10

1.567

2

14.20

10

1.874

3

11.00

10

1.333

Total

10.97

30

3.113

  •  
  •  

Multiple Comparisons

Dependent Variable:   s100number 

LSD 

(I) groups

(J) groups

Mean Difference (I-J)

Std. Error

Sig.

95% Confidence Interval

Lower Bound

Upper Bound

1

2

-6.500-*

.719

.000

-7.97-

-5.03-

3

-3.300-*

.719

.000

-4.77-

-1.83-

2

1

6.500*

.719

.000

5.03

7.97

3

3.200*

.719

.000

1.73

4.67

3

1

3.300*

.719

.000

1.83

4.77

2

-3.200-*

.719

.000

-4.67-

-1.73-

*. The mean difference is significant at the 0.05 level.

Correct grammatical errors and provide abbreviations throughout the manuscript.

Reply

  • Based on your valuable notation, the whole manuscript was revised thoroughly and all the spelling mistakes, grammar fallacies and linguistic construction were adjusted.

Provide primary references in the introduction and discussion sections.

Reply

  • The whole manuscript was revised thoroughly, and we adjusted these problems

Round 2

Reviewer 1 Report

we thank the reviewers for their efforts; this study lacks sound experimental design and this renders the results inconclusive. 

Author Response

thank you for your comments

Reviewer 3 Report

Most of concerns were solved. But those following important information are not included in the revised manuscript.  Authors should integrate them clearly into the manuscript for the readers.  

Line116: dose the oral administration here mean oral administration gavage ? what is the volume of drugs ?

  • Yes, oral administration means drug administration by oral gavage.
  • Regarding vigabatrin, the final concentration was 25mg in each one milliliter of 0.9% sodium chloride
  • Regarding empagliflozin, the final concentration was 2 mg in each one milliliter of DMSO

A work flow for experimental design is needed including all the experiments time and order.

  • We appreciate your valuable advice, a work flow for experimental design is included in the modified manuscript.

What is the tissue used for detecting SOD, MDA, mTOR, Beclin1, SIRT1, P62? cerebellum?

  • Yes, the above parameters have been detected in cerebellum.

Line 133, how to record the number of squares crossed? by video or by eye? information is needed. Whether the total travelling distance changed or not?

  • The number,of squares crossed with all paws was observed and counted manually by an observer in a 6-minutes session. The observer was blind to the animal condition.
  • Total travelling distance was not recorded in our study.

Author Response

thank you for your  valuable comments

Line116: dose the oral administration here means oral administration gavage? what is the volume of drugs?

  • The dose has been integrated clearly in the manuscript.

A workflow for experimental design is needed including all the experiments time and order.

  • The workflow for experimental design is included time and order.

What is the tissue used for detecting SOD, MDA, mTOR, Beclin1, SIRT1, P62? cerebellum?

  • We have adjusted this point clearly in the manuscript.

Line 133, how to record the number of squares crossed? by video or by eye? information is needed. Whether the total travelling distance changed or not?

We have added this data clearly in the manuscript.
